# Intercultural Differences in the Development of Pediatric Medical Traumatic Stress (PMTS) in Children Following Surgical Hospitalization

**DOI:** 10.3390/children9040526

**Published:** 2022-04-07

**Authors:** Bushra Masalha, Shiri Ben-David, Fortu Benarroch, Amichai Ben-ari

**Affiliations:** 1Herman Dana Division of Child and Adolescent Psychiatry, Hadassah-Hebrew University Medical Center, Jerusalem 91240, Israel; bushra.masalha@mail.huji.ac.il (B.M.); shiri.ben-david@mail.huji.ac.il (S.B.-D.); fortuben@hadassah.org.il (F.B.); 2Department of Psychology, Hebrew University, Jerusalem 9190501, Israel; 3Department of Psychiatry, Hadassah-Hebrew University Medical Center, Jerusalem 91240, Israel; 4Department of Behavioral Sciences, Ariel University, Ariel 9318659, Israel

**Keywords:** pediatric medical traumatic stress, post-traumatic stress disorder, cultural differences, children after hospitalization

## Abstract

Background: Illness, surgery and surgical hospitalization are significant stressors for children. Some children who experience such a medical event may develop Pediatric Medical Traumatic Stress (PMTS). PMTS affects physical recovery, and many areas and functions in children’s lives, both short- and long-term. The aim of the study is to examine the difference in the rate of PMTS between the Arab and Jewish populations and the difference in risk factors for the development of this syndrome. Method: The study involved 252 parents of children aged 1–6 who were hospitalized in the surgical ward of Hadassah Medical Center. During hospitalization, parents completed questionnaires to identify risk factors for the development of PMTS. At 3 months from the time of discharge, the children’s level of PMTS was measured. Results: The rate of children diagnosed with PMTS among Arab children was significantly higher than the rate in the Jewish population. The affiliation to an ethnic group affected different socioeconomic, demographic, social, linguistic and cultural background variables, which in turn affected the emergence of PMTS. Conclusion: The study emphasizes the nature of PMTS at the intercultural level, which can be an important source for theoretically understanding both the disorder and culture, as well as for clinical implications in developing population-sensitive treatment.

## 1. Introduction

### 1.1. Pediatric Medical Traumatic Stress (PMTS)—Description of the Phenomenon

Medical events, such as diagnosis, illness, hospitalization, surgery and medical treatment are mental stressors for children. Following a medical event, some of the children will recover spontaneously without being left with chronic mental symptoms. However, others will suffer post-traumatic symptoms that will affect their emotional, functional, and physical state [1,2,3,4,5]. An acute traumatic-stress reaction to hospitalization is frequent [6], but most children recover spontaneously. Nevertheless, 25.0–30.0% of the children will develop chronic post-traumatic symptoms affecting the course of their physical recovery and overall functioning, and 10.0–20.0% of them will meet the criteria for PTSD outlined in the fifth edition of the Diagnostic and Statistical Manual of Mental Disorders (DSM-V) [7]. The remaining children who do not meet the full diagnostic criteria for PTSD still have significant emotional distress and/or significantly disrupted functioning, which persists for a long time after hospitalization.

In recent years, awareness has risen of the mental, emotional, and behavioral consequences of medical events on children [8,9]. In order to signify the unique aspects of a condition caused by a medical event, the term pediatric medical traumatic stress (PMTS) has been used [10]. PMTS is characterized by an array of physical and emotional symptoms, such as hyper-arousal, avoidance, difficulty sleeping, nightmares, tantrums, aggression and a re-experience of the event, which have developed due to a major illness or medical intervention that involved a threat to the child’s health and involved an intrusive, painful, and frightening medical care [11,12]. Because of the low awareness of medical staff of the psychological consequences of medical events on children, many children who develop post-traumatic distress after a medical event are undiagnosed and not referred for psychiatric or psychological treatment [13,14,15].

### 1.2. Prevalence of PMTS

Stress symptoms have been found to be common in most children who have experienced medical or surgical hospitalization. In most children, the symptoms pass without professional intervention, but among a significant proportion of them, emotional and functional distress remains [6,7,12,16]. Studies in young children (0–6 years) show that the incidence of PMTS in the first month after the medical event ranges from 7.0–29.0%, 10.0% after six months, and 2.4% three years after the medical event [5]. In a study conducted by our group among children who were hospitalized for various surgical interventions in a pediatric surgery ward, about 10.4% of them developed PMTS, and about 26.4% presented post-traumatic symptoms that caused functional problems and adjustment difficulties [17,18].

### 1.3. Risk Factors for the Development of Pediatric Medical Traumatic Stress in Children after Hospitalization

Factors found to be at risk for developing PMTS in children include: the level of anxiety and worry of the child and parent [19,20,21], parental functioning [22,23], characteristics of hospitalization and medical intervention, such as the duration of hospitalization; the severity of the injury; degree of pain; nature and number of medical procedures [19,21,24,25]; characteristics of the child, such as temperament [20] and the child’s previous emotional and behavioral functioning [23,24]; familial and social factors [19,26]; and, finally, the socioeconomic and demographic status of the family [27], also among Arab children [28].

### 1.4. Post-Traumatic Symptoms in Children in Arab Society

Studies on post-traumatic stress disorder in children in Arab societies focus mainly on war events related to trauma [29,30] or cancer [31,32]. To the best of our knowledge, no studies have examined post-traumatic symptoms due to hospitalization in a surgical ward among children in Arab society. In light of this, the present study is innovative and may provide missing and necessary information for the diagnosis and treatment of children in Arab and Jewish societies.

### 1.5. Frequency of Injuries among Arab and Jewish Populations in Israel

Compared to children (ages 0–9) living in Jewish or mixed localities, children living in Arab localities are at increased risk of injury, in all age categories and at the three levels of injury (referral to emergency, hospitalization and mortality). In addition, the more severe the level of harm, the higher the proportion of Arab children [33]. Moreover, the risk of injury to a child living in an Arab locality is higher than that of a child living in a Jewish locality and increases as the severity of the injury increases [33]. Hospitalization rates for children living in Arab localities are 1.5 times higher than their rates in Jewish localities [33]. The main reasons for hospitalization are car accidents, falls and burns [34]. Possible reasons for these differences between Arab and Jewish populations in injury rates are low socioeconomic status, demographic situation, deficiencies in infrastructure in Arab localities, low level of law enforcement, the physical environment in which children live and the level of child care [34].

Differences between the two populations also refer to the degree of accessibility to a medical center which affects the degree of motivation of parents to seek medical help, as well as to the alternatives of emergency care offered [33].

### 1.6. Unique Characteristics of Hospitalized Arab Children

Avoidance symptoms were found to be the most common characteristic of hospitalized Arab children [31]. This symptom also has been found to be common in other minority populations [35]. People from Arab societies tend to express their emotional difficulties through somatic symptoms, thus enabling them not to deal with a mental disorder and the stigma associated with it [36]. They also tend not to reach out to mental health professionals and instead turn to the clergy. Exacerbation of the symptoms of mental disorders linked to belonging to a minority group correlates to the lack of awareness of mental problems (including post-traumatic stress disorder), and it also correlated with a negative stigma towards psychiatric diagnoses. That stigma might cause avoidance to seek professional treatment [37].

### 1.7. The Effect of Intercultural Variables on PTSD

Ungar [38] reviewed the relationship between factors associated with resilience, and aspects of the individual’s social ecology (environment) that promote and protect against the negative impact of exposure to traumatic events and found that resilience is sensitive to individual, contextual, and cultural diversity. A study examining post-traumatic distress among children living in war zones also found that intercultural differences are a significant factor in predicting a child’s distress level [39]. In addition, a meta-analysis study that examined variables impacting post-burn psychological adjustment found that there was a difference in the percentages of post-traumatic stress disorder among populations with different ethnic backgrounds [40].

### 1.8. Rationale and Research Hypotheses

The high rate of hospitalization among Arab children, in particular, combined with the lack of awareness of the possibility of emotional distress following hospitalization and the avoidance of seeking mental health care among the Arab population (partly due to the “stigma” problem), can cause many Arab children to suffer from PMTS after hospitalization or surgery. Due to the lack of awareness of the phenomenon, it is possible that this distress will not be identified and therefore they will not be referred for treatment and, as a result, the distress will increase. The aim of the study was to examine the difference in the rate of PMTS between Arab and Jewish populations and whether risk factors for the development of this syndrome are different in the two populations. Understanding differences between populations will allow medical staff to tailor intervention during hospitalization and allow mental health professionals to make a more accurate assessment of the need for further treatment.

## 2. Method

### 2.1. Participants

An amount of 252 children from the pediatric surgical ward at Hadassah Medical Center were included in the study: 146 boys (57.7%) and 106 girls (42.3%), and their parents—75 fathers (30.0%) and 177 mothers (70.0%). In total, 151 of the children were Jewish and Hebrew speakers (59.7%), and the rest were Arabic speakers, most of them Muslim (*n* = 97, 95.0%) and 3.0% were Christians. Some of the Arab children in the sample came from the Palestinian Authority (about 25.0%) and some from Israel (about 75.0%). The age range of the children was 1–6 years (M = 2.99, SD = 1.55). Of the children, 175 were hospitalized for elective surgery (69.2%), and the rest for emergency interventions (*n* = 78, 30.8%). Hospitalization duration ranged between 1–37 days (M = 4.71, SD = 5.70).

The ages of the Arab children (years, M = 3.12, SD = 1.60) did not significantly differ from the ages of the Jewish children (years, M = 2.89, SD = 1.52) t(250) = −1.15. Percentage of emergency surgeries among Arab children (*n* = 30, 29.0%) also did not significantly differ from that of Jewish children (*n* = 48), (31.8%). Hospitalization duration (days) was higher for Arab children (M = 7.56, SD = 7.65) than for Jewish children (M = 2.78, SD = 2.39) t(114.50) = 6.10 ***.

Demographic and hospitalization data (gender ratio, reason of admission, length of hospitalization) of the sample group were similar to those of the overall population of this age group hospitalized in the ward (*n* = 6231) during 2018. Therefore, the research sample is representative of this population.

### 2.2. Measures


The screening questionnaire (PMTSSQ) [41].


The “Pediatric Medical Traumatic Stress Screening Questionnaire” (PMTSSQ) is a screening tool for early detection of children at risk of developing PMTS. It is a questionnaire filled out by the parent for the child. The tool was built for young children (ages 1–6) who are at risk of developing PMTS after hospitalization or surgery. The questionnaire includes 32 items that examine the following variables: socio-demographic, socio-economic, parents’ level of anxiety during hospitalization, child’s anxiety level during hospitalization, parental lack of control, child’s previous psychological and behavioral functioning, parental and family functioning, level of exposure for pain and invasive interventions during hospitalization, the level of exposure to previous traumas, and the degree of exposure of the child to information about his or her medical condition. The internal reliability scale (Cronbach’s α) in the present study was α = 0.95. The tool has undergone a validation process, and the manuscript has been submitted for publication recently.
2.UCLA PTSD Reaction Index for DSM-5 Parent/Caregiver Version for Children Age 6 Years and Younger [42].

This questionnaire examines PTSD among children and adolescents. The UCLA PTSD Reaction Index questionnaire was developed in 1985. The tool was found to be effective in assessing post-traumatic stress disorder, with high validity and reliability (Cronbach’s α = 0.88–0.91), with no significant differences in scores between ethnic and religious groups [43]. The results of this measure are consistent with the results of other PTSD questionnaires, such as the PTSD Checklist, PTSD Symptom Scale, and the Harvard Trauma Questionnaire [42]. The internal reliability scale (Cronbach’s α) in the present study was α = 0.95.
3.Young Child PTSD Checklist (YCPC) [44].

This questionnaire examines PTSD among preschool children in accordance with DSM-5 criteria. This is a caregiver report on the child’s symptoms questionnaire that comprises 42 items. The first 13 items examine the type of traumatic event the child has undergone. The second section contains 23 items regarding the symptoms the child is experiencing. The last part includes 6 items about functional impairment. This questionnaire was found to have high reliability (arousal: Cronbach’s α = 0.77, avoidance: Cronbach’s α = 0.79, reliving: Cronbach’s α = 0.81) [45], indicating its effectiveness in diagnosing PTSD in preschool children. Good internal reliability for symptom scales (Cronbach’s α) was found in the current study.

arousal: α = 0.92, avoidance: α = 0.93, reliving: α = 0.88, total score: α = 0.97. Good internal traceability for the functioning scale was also found in the current study: α = 0.96.

All three research tools have been translated into Arabic specifically for the present study. The translation was produced using the “double translation” method [46].

### 2.3. Procedure

The research procedure consisted of two steps. In the first phase, all subjects gave their informed consent for inclusion before they participated in the study. The study was conducted in accordance with the declaration of Helsinki, and the protocol was approved by the ethics committee of Hadassah Medical Center (0437-14-HMO). In addition, an application was made to all parents whose children were hospitalized at the time of the research—316 parents of whom 287 agreed to participate (reasons for refusal were unwillingness and lack of time). The study was conducted during the months of April to July 2018. An explanation of the course of the study was provided to parents, and a PMTSSQ questionnaire was transferred. In the second phase, which was conducted approximately three months after discharge, the parents completed the questionnaires at UCLA, YCPC, in order to measure the development of post-traumatic symptoms in children. Out of the 287 parents who participated in the first step, 252 parents agreed to participate in the second phase (32 did not answer to repeated phone calls, and 3 refused due to a lack of time). No significant differences were found between the second stage participants and dropouts in the relevant background variables. The diagnosis of PMTS in this study was made according to PTSD diagnosis based on data from the UCLA PTSD and YCPC questionnaires.

## 3. Results

### 3.1. Group Differences

PMTS rate among Arab children (24.5%) was significantly higher compared to Jewish children (8.6%, χ^2^(1) = 12.06, *p* = 0.001). Group differences in background variables and screening questionnaire measures are presented in Table 1. As can be seen, socioeconomic difficulties and previous trauma were more frequent among Arabs as compared to Jews. Hospitalizations of Arab children were longer. Among Arabs, previous child and parent functioning were reported as lower. Arab parents reported higher parental and child anxiety levels as compared to Jews. 

### 3.2. PTSD Predictors among Arabs and Jews

Correlations between background and screening variables, measured during hospitalization, and PTSD diagnosis evaluated after discharge, among Arabs and Jews separately, are presented in Table 2 and Table 3. As can be seen, emergency surgery (compared to elective) and longer hospitalization were predictors of PTSD among both Jews and Arab. Socioeconomic problems were significantly related to PTSD among Arab children only. Among the rest of the screening measures, all were positively related to PTSD among Arabs, whereas, among Jews, only the child’s anxiety level and the parents’ perception of severity were significantly correlated to later PTSD.

### 3.3. Mediation Analyses

Background and screening variables were tested as possible mediators for the relationship between ethnic groups and PTSD. The basic mediation model is presented in Figure 1. The unstandardized path coefficients indicate the direct effects of the ethnic group on the mediators—background and screening variables (path a), the direct effects of mediators on PTSD prevalence (path b) and the direct effect of ethnic group on PTSD (path c) are presented in Table 4. The estimations of the indirect effects, including the bootstrapping CIs, are also presented in Table 4. As can be seen, previous trauma, hospitalization duration, child’s and parent’s previous functioning and the child’s anxiety level during hospitalization significantly mediated the relationship between the ethnic group and PTSD. Previous trauma mediation was partial (path c remained significant) while other mediations were full.

## 4. Discussion

To the best of our knowledge, this study is the first to examine the proportion of Arab children who suffer post-traumatic stress symptoms due to a medical event. Findings indicate that there is a significant difference in the frequency of PMTS between Arabs and Jews who are hospitalized in a surgical ward, and that there are differences in the risk factors for this frequency among the populations.

Results show that risk factors for developing PMTS in children are the child’s and parents’ previous psychological state and level of functioning, the child’s and parents’ level of anxiety during hospitalization, how the parent perceives the severity of the medical condition, and hospitalization characteristics. These variables predict PMTS three months after discharge. These findings are similar to previous studies in the field [19,20,21,22,24]. However, the present study found that, unlike Jewish children, among Arab children, the socioeconomic and demographic status of the family is a risk factor for developing PMTS. This finding is consistent with studies in the field of post-traumatic stress disorder (PTSD) among children of different backgrounds [27] and among Arab children in particular [28]. As in other studies [47,48,49], no association was found between rates of PTSD and child gender, child and parental age, and the child’s position in the family. 

According to the study results, 8.6% of the Jewish children and 24.5% of the Arab children were found to suffer from PMTS three months after discharge from hospitalization in the surgical ward. This difference is partially explained by the length of hospitalization and the severity of the medical condition characterizing Arab children [33]. From these data, it appears that in early childhood, Arab children are highly vulnerable to developing PMTS. This is in line with the literature showing that belonging to a minority group and socioeconomic and demographic difficulties are associated with the risk of experiencing post-traumatic stress disorder [27,28,35,50,51,52]. This fact allows us to hypothesize that the high prevalence of PMTS among Arab children is due to background factors and lower socioeconomic status.

Another explanation for the difference between Arab and Jewish children is related to cultural gaps and language difficulties. Some of the Arab children in the sample came from the Palestinian Authority (about 25.0%) and some from Israel (about 75.0%). Arabs (both from PA and Israel) encounter difficulties arising from language limitations and cultural and social differences. It was found that the Israeli health system lacks culturally competent treatment [53] and that there are health disparities between Hebrew speakers and Arabic speakers [54,55,56]. In addition, Arabs encounter difficulties in receiving adequate health care services, due to language limitations and a lack of awareness among medical staff regarding the unique needs and cultural characteristics of this population [33,57]. The Arabic population tend to have a negative stigma regarding mental difficulties and psychopathology [37], which makes it more difficult for professionals to identify and offer treatment, while time plays a significant role in the recovery process [4,11,58].

The present study has several limitations. Some of the children diagnosed in the current study with PMTS have experienced a traumatic event in the past (Arab children more than Jewish children). Hence, it is possible that the main traumatic event of some of the children is related to their personal history and not necessarily to the medical event that occurred prior to hospitalization. It has been found that a child’s previous traumatic background affects his or her risk level of developing post-traumatic stress disorder from another event [59]. However, it should be noted that all parents of children diagnosed with PMTS in the study indicated that hospitalization was the main traumatic event for them (even if they had previous traumatic events). Another limitation has to do with the way data was collected, which relied on parental reporting. Since the children are under 6 years old, it was difficult to get a direct report from them about their condition and distress. Yet, it is acceptable to get information about the mental state of preschool children through parental reports [60]. However, in order to increase the validity of the report, a number of questionnaires were used to measure distress among children, which reinforced the results. It was also found that parents’ assessment of post-traumatic stress symptoms for their child is highly correlated with children’s reports of themselves [61]. Another limitation of the study is that information on parental PTSD was not collected and it is possible that parental distress symptoms may skew their reporting of their child’s symptoms. Another limitation in the study is that PTSD diagnosis was made based on questionnaires, which is not valid as a diagnosis made by clinical interview.

It should be noted that 25.0% of the children in the sample came from the Palestinian Authority because their difficult medical conditions did not allow them to receive an appropriate medical response in Palestinian Authority hospitals. It could be argued that due to the fact that the less complicated medical conditions are not treated in Israel, the sample might be biased. However, in fact, most of the Arab children are from Israel (75.0%) while just 25.0% are from the Palestinian Authority. Arab children from Israel also suffer from more severe injuries and lack of access to a medical center [33].

### The Significance of the Findings and Their Theoretical and Clinical Implications

For the first time in the literature, our study indicates the outstanding percentage of PMTS cases among Arab children, which demands more awareness to promote prevention, early detection and treatment. Another significant finding shows that the PMTSSQ questionnaire optimally detects Arab children with a risk of developing PMTS, since the risk factors that it reviews are relevant to the Arab population, and therefore, it can be used among Arab children. Our findings indicate that there is a relationship between the level of PMTS and socioeconomic status and background factors among Arab children as compared to Jewish children. The affiliation to an ethnic group also affects the background and screening variables, which in turn affect the onset of post-traumatic symptoms, thus Arabs come with different socioeconomic, demographic, social, linguistic and cultural background variables that put them at greater risk for developing post-traumatic stress disorder following a medical incident.

The fact that the study was conducted among Arab and Jewish populations allows for intercultural learning. Comparative studies regarding the tool and nature of the disorder at the intercultural level can be an important source of learning, both for the disorder and for the culture, both theoretically and therapeutically, in what characterizes PMTS among different populations.

Clinically, the study highlights the need to develop a predictive tool sensitive to language and culture in order to meet the psychosocial needs of different ethnic groups in medical hospitalization situations, as well as the awareness to look for specific symptoms among the Arab society as a minority [51]. It is worth considering using the services of medical interpreters, thus enabling access to medical information, providing explanatory forms, and respecting differences in attitudes and approaches, especially when the clinical condition is sensitive and complex [62]. It is recommended that medical staff be trained in intercultural therapy skills based on an approach of culturally tailored and sensitive therapy that allows patients to be approached as individuals and not generalized on the basis of general ethnic groups [37,63].

Eventually, a change in policy and practice will lead to an improvement in the emotional and mental state of patients arriving from different ethnic populations. It will also help to guide psychotherapists to become culturally-adapted and symptom-adapted according to the patient’s ethnic origin [35].

## Figures and Tables

**Figure 1 children-09-00526-f001:**
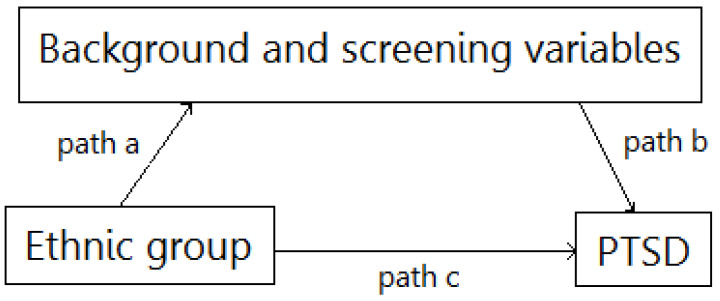
Mediation model.

**Table 1 children-09-00526-t001:** Group differences.

	Jews (*n* = 151)	Arabs (*n* = 102)	Group Differences
Age of child (years, M [SD])	2.89 (1.52)	3.12 (1.60)	t(250) = −1.15
SES (difficulties, *n* [%])	32 (22.4%)	47 (46.1%)	χ ^2^(1) = 15.31 ***
Previous trauma (M [SD])	1.17 (0.61)	1.63 (1.35)	t(129.01) = 3.20 **
Emergency surgery (*n*, %)	48 (31.8%)	30 (29.4%)	χ^2^(1) = 0.16
Hospitalization duration (days, M [SD])	2.78 (2.39)	7.56 (7.65)	t(114.50) = 6.10 ***
Child’s previous functioning ^ (M [SD])	1.43 (0.51)	1.92 (1.25)	t(123.67) = 3.71 ***
Parental previous functioning ^ (M [SD])	1.55 (0.70)	2.08 (1.25)	t(143.21) = 3.92 ***
Child’s anxiety level (M [SD])	2.07 (1.01)	2.76 (1.58)	t(156.72) = 3.93 ***
Parental perception of severity (M [SD])	2.53 (1.07)	2.79 (1.41)	t(176.18) = 1.59

** *p* < 0.01, *** *p* < 0.001. ^ High levels represents problematic functioning, negative beliefs.

**Table 2 children-09-00526-t002:** Correlations between PTSD and background and screening variables—Arabs (*n* = 102).

	1	2	3	4	5	6	7	8	9
1. PTSD	-								
2. Age of child	−0.04	-							
3. SES	0.30 **	0.05	-						
4. Previous trauma	0.62 ***	0.02	0.33 **	-					
5. Surgery type	0.68 ***	−0.07	0.14	0.44 ***	-				
6. Hospitalization duration	0.69 ***	−0.06	0.20 *	0.39 ***	0.49 ***	-			
7. Child’s previous functioning	0.68 ***	0.06	0.34 ***	0.50 ***	0.41 ***	0.48 ***	-		
8. Parental previous functioning	0.72 ***	0.03	0.42 ***	0.62 ***	0.51 ***	0.50 ***	0.89 ***	-	
9. Child’s anxiety level	0.71 ***	−0.10	0.34 ***	0.49 ***	0.56 ***	0.52 ***	0.75 ***	0.76 ***	-
10. Parental severity perception	0.79 ***	−0.09	0.31 **	0.54 ***	0.60 ***	0.62 ***	0.81 ***	0.78 ***	0.87 ***

* *p* < 0.05, ** *p* < 0.01, *** *p* < 0.001.

**Table 3 children-09-00526-t003:** Correlations between PTSD and background and screening variables—Jews (*n* = 151).

	1	2	3	4	5	6	7	8	9
1. PTSD	-								
2. Age of child	0.03	-							
3. SES	0.14	0.00	-						
4. Previous trauma	0.15	0.07	0.07	-					
5. Surgery type	0.25 **	−0.07	0.01	0.04	-				
6. Hospitalization duration	0.35 ***	−0.09	0.06	−0.05	0.18 *	-			
7. Child’s previous functioning	0.17 *	0.17 *	0.13	0.21 *	−0.05	0.00	-		
8. Parental previous functioning	0.16	0.01	0.25 **	0.02	−0.01	0.08	0.09	-	
9. Child’s anxiety level	0.14	−0.12	0.06	0.05	0.32 ***	0.16	0.18 *	0.02	-
10. Parental severity perception	0.26 ***	−0.12	0.16	0.21 **	0.38 ***	0.20 *	0.22 **	0.03	0.29 ***

* *p* < 0.05, ** *p* < 0.01, *** *p* < 0.001.

**Table 4 children-09-00526-t004:** Mediation models for the relationship between ethnic group and PTSD.

Mediator	Path a		Path b		Path c		Indirect Effect (Bootstrapping)	
	B (SE)	t	B (SE)	z	B (SE)	z	B (SE)	95% CI
SES								
Previous trauma	−0.45 (0.12)	−3.64 ***	0.90 (0.17)	5.29 ***	−0.80 (0.41)	−1.92 *	−0.41 (0.17)	−0.78; −0.14
Surgery type								
Hospitalization duration	−4.79 (0.67)	−7.15 ***	0.30 (0.005)	5.72 ***	0.25 (0.49)	0.51	−1.43 (0.40)	−2.38; −0.82
Child’s previous functioning	−0.48 (0.11)	−4.27 ***	1.30 (0.22)	5.95 ***	−0.33 (0.46)	−0.73	−0.63 (0.21)	−1.10; −0.28
Parental previous functioning	−0.53 (0.12)	−4.35 ***	1.25 (0.20)	6.28 ***	−0.44 (0.45)	−0.99	−0.67 (0.21)	−1.14; −0.31
Child’s anxiety level	−0.69 (0.16)	−4.26 ***	1.18 (0.20)	5.97 ***	−0.20 (0.47)	−0.43	−0.82 (0.26)	−1.42; −0.40
Parental severity perception	−0.26 (0.16)	−1.68	1.47 (0.22)	6.72 ***	−0.91 (0.46)	−1.96 *	−0.39 (0.26)	−0.99; 0.05

* *p* ≤ 0.05, *** *p* < 0.001. SE=standard error. SES=socioeconomic status.

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
