# Peer review of "Intercultural Differences in the Development of Pediatric Medical Traumatic Stress (PMTS) in Children Following Surgical Hospitalization"

_children, 2022, doi:10.3390/children9040526_

Round 1

Reviewer 1 Report

Review

Thank you for the opportunity to review this work. This study sought to examine intercultural differences, comparing Arab and Jewish preschool children regarding frequency of Pediatric Medical Traumatic Stress (PMTS) and the mediators of PMTS following surgical hospitalization. Due to limited knowledge on intercultural differences regarding PMTS and high vulnerability of Arab children, this study seems timely in the research field, with important clinical implications. I have a few points that may strengthen the manuscript and improve the information clinicians may get from this study. Manuscript is overall well organized.

Minor points:

Line 70-74: Please replace periods after citation with commas.

Please report percentage in the whole manuscript consistently, you reported either without or with two decimal places.

Lines 137 and 138: Please close the second bracket.

Lines 176-178: Are the reported reliabilities found in your study sample? If so, please make it clear in the text.

Line 320: Please remove another period at the end of the sentence. 

Major points:

Line 148: Since parents reported on the child´s symptoms and its own symptoms, it is not proper to state that PMTSSQ is a self-report questionnaire. Please remove “self-report”. Furthermore, in the Procedure section you stated that PMTSSQ is an interview administered by clinicians. Please correct this inconsistence while describing the instrument.  

Line 161: UCLA PTSD Reaction Index for DSM-5 is developed to assess PTSD symptoms in children and adolescents, not adults. Please replace “adults” with “adolescents”.

Line 170: YCPC is not parental self-report questionnaire, it is caregiver report on the child´s symptoms. Please correct is.

Line 191-193: Please compare families, which completed the questionnaires at the second measurement point with families that dropped out from the study, regarding relevant variables collected at baseline. This is important due to generalizability of your findings.

Results: It is not clear how you operationalized PMTS? Is it a PTSD diagnosis based on questionnaire data, or did you use certain cut-offs for UCLA and YCPC? Please complete this information in Measures and Procedure sections. What was the procedure if only one questionnaire suggested PMTS, but not the other?

Please note in limitations that there is no information on parental PTSS/PTSD. It is well known from the literature that child´s exposure to traumatic event may precipitate mental health problems in their parents, including PTSD. Furthermore, parental symptoms may bias their report on their child´s symptoms.

Please note in limitations that PTSD diagnosis was made based on questionnaires, which is not valid as diagnosis made by clinical interview.      

Author Response

We thank the reviewer for the important comments. The following is a reference to each of the points raised:

Point 1: Line 70-74: Please replace periods after citation with commas.

Response 1: The comment has been corrected

Point 2: Please report percentage in the whole manuscript consistently, you reported either without or with two decimal places.

Response 2: The comment has been corrected. All percentages in the article appear uniformly with one decile beyond the dot.

Point 3: Lines 137 and 138: Please close the second bracket.

Response 3: The comment has been corrected.

Point 4: Lines 176-178: Are the reported reliabilities found in your study sample? If so, please make it clear in the text.

Response 4: The comment has been corrected. It was clarified that the reported reliability was found in the present study.

Point 5: Line 320: Please remove another period at the end of the sentence.

Response 5: The comment has been corrected

Point 6: Line 148: Since parents reported on the child´s symptoms and its own symptoms, it is not proper to state that PMTSSQ is a self-report questionnaire. Please remove “self-report”. Furthermore, in the Procedure section you stated that PMTSSQ is an interview administered by clinicians. Please correct this inconsistence while describing the instrument.

Response 6: The comment has been corrected

Point 7: Line 161: UCLA PTSD Reaction Index for DSM-5 is developed to assess PTSD symptoms in children and adolescents, not adults. Please replace “adults” with “adolescents”.

Response 7: The comment has been corrected

Point 8: Line 170: YCPC is not parental self-report questionnaire, it is caregiver report on the child´s symptoms. Please correct is.

Response 8: The comment has been corrected

Point 9: Line 191-193: Please compare families, which completed the questionnaires at the second measurement point with families that dropped out from the study, regarding relevant variables collected at baseline. This is important due to generalizability of your findings.

Response 9: The comment has been corrected. If the average table is needed, we can send it later.

Point 10: Results: It is not clear how you operationalized PMTS? Is it a PTSD diagnosis based on questionnaire data, or did you use certain cut-offs for UCLA and YCPC? Please complete this information in Measures and Procedure sections. What was the procedure if only one questionnaire suggested PMTS, but not the other?

Response 10: The diagnosis of PMTS was made according to PTSD diagnosis based on data from the UCLA PTSD and YCPC questionnaires. All children diagnosed in the PMTS study met the PTSD criteria in both questionnaires and therefore there was no need to decide between the two as there were no cases of discrepancy. The reason we are talking about PMTS in the article and not PTSD is because it is exacerbated by post-trauma symptoms with a medical context. However, following this important remark, the point in the procedure chapter has been corrected.

Point 11: Please note in limitations that there is no information on parental PTSS/PTSD. It is well known from the literature that child´s exposure to traumatic event may precipitate mental health problems in their parents, including PTSD. Furthermore, parental symptoms may bias their report on their child´s symptoms.

Response 11: This is a very correct remark and we have added a reference to this point in the discussion chapter.

Point 12: Please note in limitations that PTSD diagnosis was made based on questionnaires, which is not valid as diagnosis made by clinical interview.    

Response 12: This is also a very important comment and we also addressed it at this point in the discussion chapter.

Reviewer 2 Report

The manuscript entitled “Intercultural differences in the development of Pediatric Medical Traumatic Stress (PMTS) in children following surgical hospitalization” by Masalha et al. focused on the nature of PMTS at the intercultural level important for understanding the disorder in developing population-sensitive treatment.

The  article is of interest to wide-range of readers. The concept and methodology is logically presented.

The studies reported on ethnic groups in addition to Jewish and Arab children should be discussed .

Author Response

Response to Reviewer 2 Comments

We thank the reviewer for the important note. The following is a reference to a point raised in the review:

Point 1: The studies reported on ethnic groups in addition to Jewish and Arab children should be discussed

Response 1: This is an important note. We have added a paragraph addressing intercultural differences among post-traumatic children in the literature review chapter.

Round 2

Reviewer 1 Report

This study sought to examine intercultural differences, comparing Arab and Jewish preschool children regarding frequency of Pediatric Medical Traumatic Stress (PMTS) and the mediators of PMTS following surgical hospitalization. Due to limited knowledge on intercultural differences regarding PMTS and high vulnerability of Arab children, this study seems timely in the research field, with important clinical implications. Thank you for addressing most of the raised points, which improved the manuscript significantly. However, there are still a few points that may strengthen the manuscript.

Author Response

Minor points:
All points have been corrected according to the comments.
Major point:
In fact there were no gaps between the questionnaires. All children who were found to have a PTSD diagnosis in the YCPC questionnaire were also found to have a diagnosis in the UCLA PTSD questionnaire.
In addition, reference was made to the subjects' consent to participate in the study and to the Helsinki Committee's ethics approval in the procedure chapter.
